# Variation Analysis of Root System Development in Wheat Seedlings Using Root Phenotyping System

**Ekundayo Adeleke [1], Reneth Millas [1], Waymon McNeal [1], Justin Faris [2] and Ali Taheri [1,*]**

[1] Department of Agricultural and Environmental Sciences, Tennessee State University, Nashville, TN 37209, USA; eadeleke@Tnstate.edu (E.A.); renz_millas@yahoo.com (R.M.); waymon.mcneal28@gmail.com (W.M.)

[2] USDA-ARS Cereal Crops Research Unit, Edward T. Schafer Agricultural Research Center, 1616 Albrecht BLVD N., Fargo, ND 58102, USA; Justin.Faris@ars.usda.gov

[*] Correspondence: ataheri1@tnstate.edu or atahery@gmail.com; Tel.: +1-(615)-963-6056

**Abstract:** Root system architecture is a vital part of the plant that has been shown to vary between species and within species based on response to genotypic and/or environmental influences. The root traits of wheat seedlings are critical for their establishment in soil and evidently linked to plant height and seed yield. However, plant breeders have not efficiently developed the role of RSA in wheat selection due to the difficulty of studying root traits. We set up a root phenotyping platform to characterize RSA in 34 wheat accessions. The phenotyping pipeline consists of the germination paper-based moisture replacement system, image capture units, and root-image processing software. The 34 accessions from two different wheat ploidy levels (hexaploids and tetraploids), were characterized in ten replicates. A total of 19 root traits were quantified from the root architecture generated. This pipeline allowed for rapid screening of 340 wheat seedlings within 10 days. At least one line from each ploidy (6× and 4×) showed significant differences ($p < 0.05$) in measured traits, except for mean seminal count. Our result also showed a strong correlation (0.8) between total root length, maximum depth and convex hull area. This phenotyping pipeline has the advantage and capacity to increase screening potential at early stages of plant development, leading to the characterization of wheat seedling traits that can be further examined using QTL analysis in populations generated from the examined accessions.

**Keywords:** root system architecture; high-throughput phenotyping; root traits; *Triticum* sp.; germination paper-based system

## 1. Introduction

Roots serve as boundaries between plants and complex soil mediums. Aside from anchoring the plant to soil medium [1], another major function of the root is to provide plant access to nutrient and water uptake. Roots are also essential for forming symbioses with beneficial microbes in the rhizosphere and used as storage organs [1,2]. Therefore, roots are critical in the maintenance of plant health. Many environmental factors interact with soils, leading to the spatial and temporal heterogenous nature of the soil [3]. This spatial heterogeneity makes studying the roots in soil a multifaceted challenge. The spatial distribution of roots in soil under field conditions demonstrates a considerable amount of variability, since roots respond to heterogeneity in the soil and environmental cues allowing plants to overcome challenges posed by biotic or abiotic factors in soil environment [2]. This spatial distribution of the root system in soil is referred to as root system architecture (RSA). RSA usually describes the morphological and structural organization of the root [4]. RSA is important for plant productivity because it determines the plant's ability to successfully access major heterogenous edaphic resources [5]. Therefore, RSA has a direct influence on grain yield.

Wheat is a major cereal crop of global importance. It is grown in temperate zones and has remained a worldwide staple food [6]. It belongs to the *Triticum* genus, which includes species such as *T. aestivum* ssp. *aestivum* L. (common wheat, $2n = 6x = 42$, AABBDD genomes), an allohexaploid and the most cultivated wheat species in the world, accounting for 95% of global wheat production [7]; *T. turgidum* ssp. *durum* (Desf.) Husnot (durum wheat, $2n = 4x = 28$, AABB genomes), a tetraploid that is the second most cultivated wheat species accounting for 5%-8% of global wheat production [8]; and *T. turgidum* ssp. *dicoccum* (Schrank) Schübl (cultivated emmer wheat, $2n = 4x = 28$, AABB genomes) a tetraploid that is one of the earliest crops domesticated in the Near East [9]. So far, most wheat breeding programs have focused on aboveground phenotypic traits while ignoring the belowground traits. Although it is easier for breeders to consider aboveground traits because they are the most visible to the eye, belowground traits should not be ignored because they play equally important roles in plant productivity [1,2].

In cereal grains, the radicle emerges first and is covered with a protective sheath called the coleorhiza [10,11]. After the roots have extended somewhat further, the coleoptile emerges and grows rapidly. The seedling will then possess a unique RSA [12] by the time they are at the germination stage (Figure 1), and this has a major impact on the early establishment of the seedling and its productivity at later growth stages.

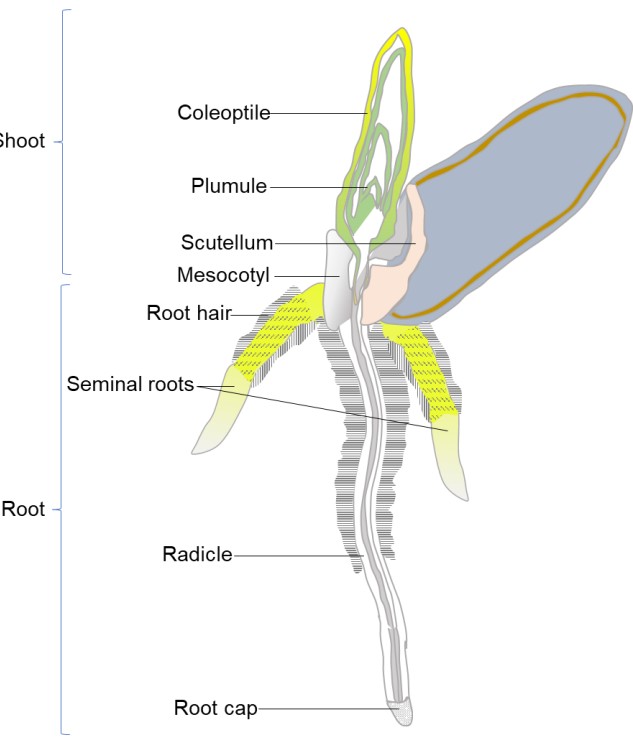

**Figure 1.** Annotated diagram of germinating 4 day old wheat grain. The kernel is showing root development that includes root cap, radicle, seminal roots, and root hairs; and shoot development that includes mesocotyl, plumule, and coleoptile at Zadok's growth stage 07 [13].

For wheat to grow and produce enough yield, it is important to understand and select unique traits in RSA as well, using aboveground traits. Abiotic stresses due to climate change have affected wheat productivity by limiting the uptake of nutrients and water [5]. This is one reason that progress in obtaining wheat varieties with increased yields has been hindered [14,15]. One way to alleviate the adverse effects of these factors on wheat yield is to select unique traits and manipulate the underlying genes associated with wheat RSA so as to optimize the water and nutrient uptake. Although root phenotyping is critical for optimizing RSA in crops, the study of roots in the field is still in its infancy. The traditional techniques used for studying roots in the field including soil coring, trenching, or

shovelomics [16,17]. Most of these techniques involve the excavation of the roots, washing off the soil on a sieve and afterward quantifying the root traits. These methods are time consuming, labor intensive, low-throughput and not efficient for genetic studies. In recent years, the use of soil-less media, hydroponics, semi-hydroponics or gel-based media have been used to study root development [18,19]. Current advancements in software development for imaging, automation, and robotics have increased the possibility of high-throughput, non-invasive studies of roots [20]. The germination paper-based approach (growth pouch) has been used to measure axile and lateral roots (Figure 2B) of maize [21] and has been recently modified for high-throughput measurement in rapeseed, barley [22], and common bean [23]. The germination paper-based approach developed by Hund et al. [21] required a plastic covering that stuck to the root and required intervention to remove the covering [24]. Although modified forms of this growth pouch have been reported [12,24–26], there is still a knowledge gap yet to be filled in the development of root phenotyping systems and high-throughput screenings of wheat. The wheat accessions selected for this study have never been reported for root variation analysis to the best of our knowledge. They are unique and represent a diverse pool of collection from different origins representing five continents (North America, South America, Europe, Africa, and Asia) (Table 1).

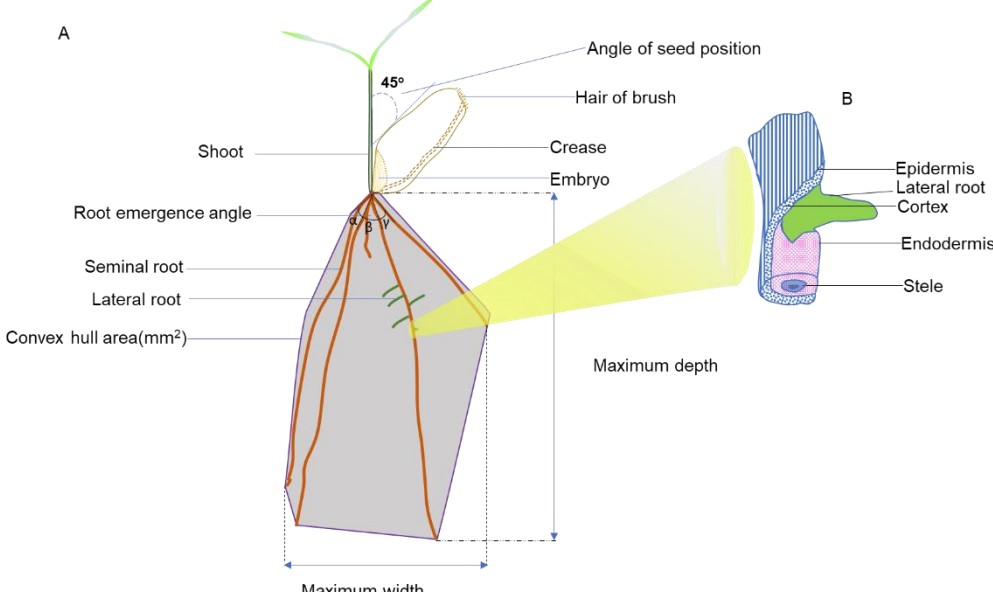

**Figure 2.** (**A**) Summary of seedling features within the growth system and the positioning of the seed. The positioning of the seed at 45° to the vertical plane of the growth system permitted the precise upward development of the coleoptile and concomitant downward growth of the roots. The crease of the seed is inverted to face the horizontal plane of the pouch, allowing the roots to grow away from the germination paper. (**B**) Illustration of lateral root emerging from the overlying tissues of the primary seminal root.

High-throughput screening can expedite the selection of novel traits for crop improvement in plant breeding [15]. However, high-throughput screening of root traits is often limited by the lack of suitable phenotyping growth systems [27]. Therefore, the main objective of this study was to evaluate variation in the RSA of seedlings from 34 wheat accessions for different root traits using a high-throughput root phenotyping pipeline.

## 2. Materials and Methods

Phenotyping of the 34 wheat accessions was divided into three stages, first, setting up the experiment on the platform; second, the acquisition of RSA images; and third, the analysis of acquired images using open source software (RootNav) [28] (Figure 3).

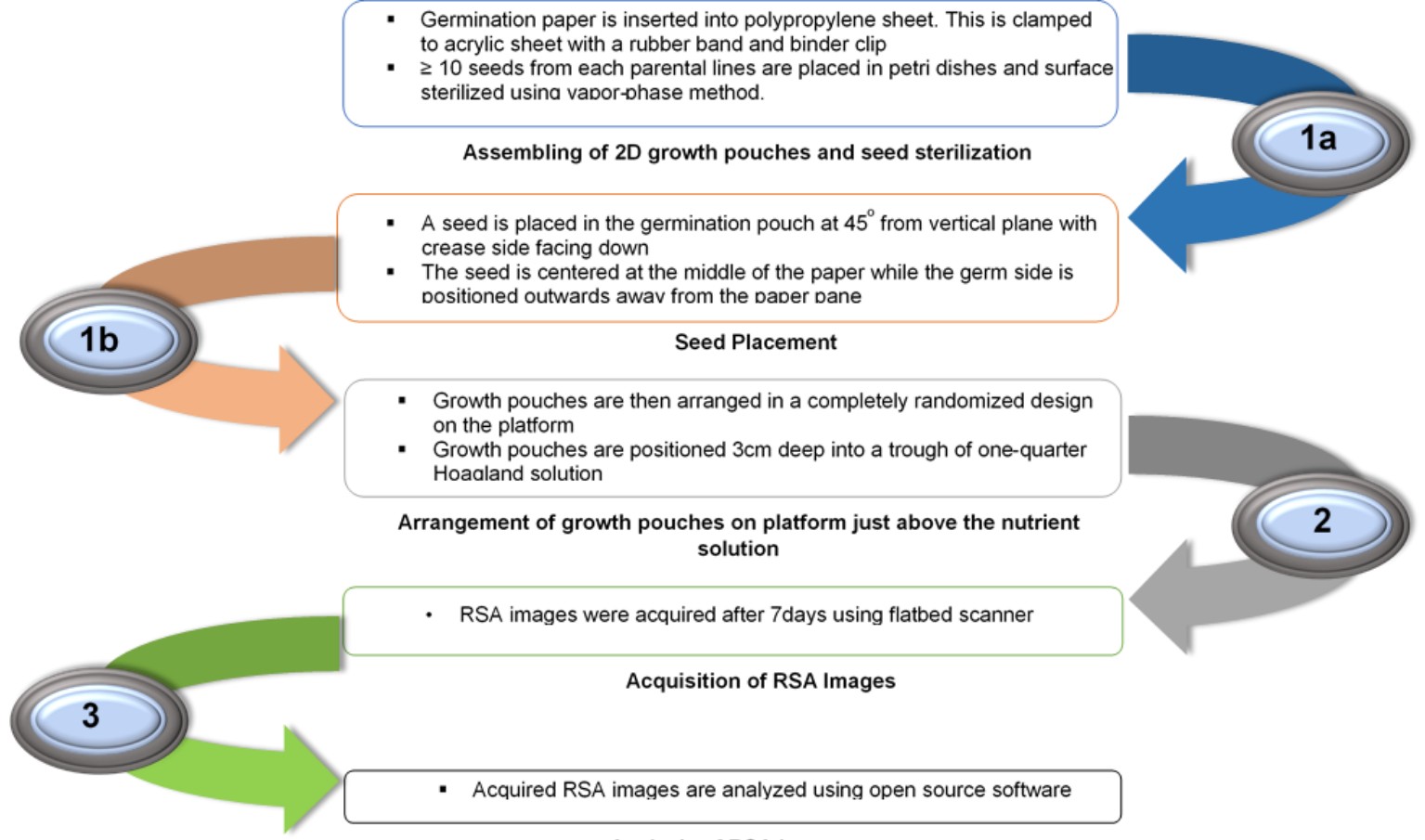

**Figure 3.** Schematic illustrating the three major steps of the root phenotyping pipeline. The first step is seed sterilization and the assembling of 2D growth pouches (1a), and the placement of seeds in respective pouches accordingly and placement into the tanks (1b). The second step involves the acquisition of RSA images using a flatbed scanner (2). The third step is the analyses of RSA images acquired in the second step (3).

The 34 accessions from different wheat species were obtained from a USDA-ARS cereal crop research unit (Fargo, ND, USA) and divided into two separate groups based on their ploidy level (hexaploid vs. tetraploid) (Table 1). The hexaploid category was made up of common wheat, spelt wheat and synthetic hexaploid wheat (SHW). SHW is hexaploid wheat that is created artificially by the introduction of additional genetic resources from tetraploid and diploid relatives to develop wheat with a broader genetic basis. SHW lines are quite useful in introducing agronomically needed traits into common wheat from wild genetic relatives [29]. In this experiment, SHW lines were selected for root phenotyping with the accession Largo selected as the reference accession based on SHW biomass uniqueness and density [29]. The tetraploid group of accessions consisted of durum (*T. turgidum* ssp. *durum*), Persian (*T. turgidum* ssp. *carthlicum*), cultivated emmer (*T. turgidum* ssp. *dicoccum*) and wild emmer (*T. turgidum* ssp. *dicoccoides*) wheat. For the tetraploid group, the durum line Rusty was selected as the reference accession.

**Table 1.** The common name, taxonomy, origin, and source of 34 different accessions assessed for its seedlings root system architecture (RSA).

| Accession | PI/CItr | Common Name | Taxon | Subspecies | Ploidy | Origin |
|---|---|---|---|---|---|---|
| Largo | CItr 17895 | Synthetic hexaploid wheat | *Triticum turgidum × Aegiliops tauschii* | Synthetic | 6× | U.S., North Dakota |
| ND495 | N/A | Common wheat | *Triticum aestivum* | aestivum | 6× | U.S., North Dakota |
| Grandin | PI 531005 | Common wheat | *Triticum aestivum* | aestivum | 6× | U.S., North Dakota |
| BR34 | N/A | Common wheat | *Triticum aestivum* | aestivum | 6× | Brazil |
| Chinese Spring | CItr 14108 | Common wheat | *Triticum aestivum* | aestivum | 6× | China |
| Arina | N/A | Common wheat | *Triticum aestivum* | aestivum | 6× | Switzerland |
| Forno | N/A | Common wheat | *Triticum aestivum* | aestivum | 6× | Switzerland |
| Sumai 3 | PI 481542 | Common wheat | *Triticum aestivum* | aestivum | 6× | China |
| Chinese Spring-DIC 5B | N/A | Common wheat | *Triticum aestivum* | aestivum | 6× | U.S., Missouri |
| Bobwhite | PI 520554 | Common wheat | *Triticum aestivum* | aestivum | 6× | Mexico, CIMMYT |
| Salamouni | PI 182673 | Common wheat | *Triticum aestivum* | aestivum | 6× | Lebanon |
| Katepwa | N/A | Common wheat | *Triticum aestivum* | aestivum | 6× | Canada |
| M3 | N/A | Synthetic hexaploid wheat | *Triticum turgidum × Aegiliops tauschii* | Synthetic | 6× | Mexico, CIMMYT |
| PI277 | PI 277012 | Spelt wheat | *Triticum aestivum* | spelta | 6× | Spain |
| M6 | N/A | Synthetic hexaploid wheat | *Triticum turgidum × Aegiliops tauschii* | Synthetic | 6× | Mexico, CIMMYT |
| Kulm | PI 590576 | Common wheat | *Triticum aestivum* | aestivum | 6× | U.S., North Dakota |
| Opata85 | PI 591776 | Common wheat | *Triticum aestivum* | aestivum | 6× | Mexico, CIMMYT |
| TA4152-60 | N/A | Synthetic hexaploid wheat | *Triticum turgidum × Aegiliops tauschii* | Synthetic | 6× | Mexico, CIMMYT |
| TA4152-19 | N/A | Synthetic hexaploid wheat | *Triticum turgidum × Aegiliops tauschii* | Synthetic | 6× | Mexico, CIMMYT |
| P503 | N/A | Spelt wheat | *Triticum aestivum* | spelta | 6× | Iran |
| Divide | N/A | Durum wheat | *Triticum turgidum* | durum | 4× | U.S., North Dakota |
| Rusty | PI 639869 | Durum wheat | *Triticum turgidum* | durum | 4× | U.S., North Dakota |
| Ben | N/A | Durum wheat | *Triticum turgidum* | durum | 4× | U.S., North Dakota |
| Lebsock | N/A | Durum wheat | *Triticum turgidum* | durum | 4× | U.S., North Dakota |
| Langdon | N/A | Durum wheat | *Triticum turgidum* | durum | 4× | U.S., North Dakota |
| Altar84 | N/A | Durum wheat | *Triticum turgidum* | durum | 4× | Mexico, CIMMYT |
| PI193 | PI 193833 | Cultivated emmer | *Triticum turgidum* | dicoccum | 4× | Ethiopia |
| PI410 | PI 41025 | Cultivated emmer | *Triticum turgidum* | dicoccum | 4× | Russia |
| PI947 | PI 94749 | Persian wheat | *Triticum turgidum* | carthlicum | 4× | Georgia |
| PI481 | PI 481521 | Wild emmer | *Triticum turgidum* | dicoccoides | 4× | Israel |
| PI478 | PI 478742 | Wild emmer | *Triticum turgidum* | dicoccoides | 4× | Israel |
| TA106 | N/A | Wild emmer | *Triticum turgidum* | dicoccoides | 4× | Israel |
| Israel A | N/A | Wild emmer | *Triticum turgidum* | dicoccoides | 4× | Israel |
| PI272 | PI 272527 | Cultivated emmer | *Triticum turgidum* | dicoccum | 4× | Hungary |

### 2.1. Experimental Design and Seed Treatment

Each accession was planted in ten replicates in a completely randomized design. The seeds were surface sterilized in a chemical hood (Labconco Inc., MO, USA) using the chlorine gas (vapor-phase) method used by Clough and Bent [30]. Ten seeds (or more) were placed in open Petri dishes (previously labeled with chlorine resistant markers) in a 10L desiccator jar. A 3 ml aliquot of 12N HCl was added to a 250 mL beaker containing 100 mL of 8.3% sodium hypochlorite before sealing the desiccator. The seeds remained in the desiccator for 4 h.

### 2.2. Design of Experimental Platform

A schematic illustration of the stages and flow of the experimental system is presented in Figure 3. We developed a growth pouch system based on the earlier platform designed by Hund et al. [21] for maize. Each sterilized seed was placed into a germination paper pouch, that was constructed from blue germination paper (21.6 × 28 cm; Anchor Paper Company, St Paul, MN, USA) inserted into Staples® standard clear polypropylene sheet protectors (Staples Inc, MA, USA) (Figures 3 and 4A). The bottom edges of these sheet protectors were removed to allow for capillary movement of distilled water and nutrient solution up the germination papers. Two germination pouches were then firmly held to either side of a clear stiff acrylic plate (0.5 × 24 × 30 cm; Acme Plastic Woodland Park, NJ, USA) with a rubber band and a binder clip (Staples Inc, MA, USA) (Figure 4A). The acrylic plates also had extended overhangs (0.5 × 1.5 × 1.0 cm) that fit into a metal support frame that was situated in the top of a customized black polypropylene tank (54.5 × 42.5 × 6.0 cm) (Figure 4C). The 2-D growth systems hung so that they were positioned about 3 cm deep into the liquid media within the tank (Figure 4B). The liquid solution consisted of 12 L of distilled water that was interchanged with modified one-quarter Hoagland's solution [31] three days after germination using a pump. The composition of the nutrient solution was 1.25 mM $KNO_3$; 0.625 mM $KH_2PO_4$; 0.5 mM $MgSO_4$; 0.5 mM $Ca(NO_3)_2$; 17.5 μM $H_3BO_3$; 5.5 μM $MnCl_2$; 0.5 μM $ZnSO_4$; 0.062 μM $Na_2MoO_4$; 2.5 μM $NaCl_2$; 0.004 μM $CoCl_2$; and 12.5 μM Fe-EDTA. The final pH of the nutrient solution was adjusted to pH 6.2.

A single seed from each accession was placed into a germination paper pouch at 2.5 cm below the top, with the crease-side down, at about a 45° orientation from the vertical plane (Figure 4A). Positioning the seed at this angle provided two main benefits. First, it allowed the phototrophic response of the coleoptile to align with the vertical plane without rerouting its mesocotyl. Second, the position also benefitted the seedling RSA by supporting root emergence away from the germination paper, resulting in easier image acquisition. Each germination pouch containing two seeds (one on each side of the acrylic plate) was arranged on the phenotyping platform in a growth room that was fitted with a growth lamp set at photoperiod of 14 h light and 10 h dark with 400 μmol $m^{-2}$ $s^{-1}$ flux density at 22 °C. After 7 days, with almost all seedlings at growth stage 10 [21], each growth pouch was removed from the platform, the polypropylene sheets were cut open on one side, and a side of each sheet was carefully opened to reveal the blue germination paper.

### 2.3. Imaging and Analysis

Imaging of the roots was carried out using a Flatbed scanner (HP Inc, Spring, TX, USA). The acquired images were saved as standardized compressed image formats (JPG files), which were then imported as new files into the RootNav software. Each image is then converted to a probability map (inverted images) in the software, with the root images represented as clustered groups of pixels using the gaussian mixture model based on the varying intensities of the pixels [28]. The RootNav allows expectation maximization clustering to assign the best appearance likelihood of the pixels from root images against the background, creating a model that can be fit from the seed point (source) to the root apices.

The RSA images acquired from the wheat seedling were then semi-automatically measured with open source RootNav software [28], and the predefined model setting for wheat seedling was used to acquire measurements of the traits. The root traits that were measured for each replicate included: total

length (TL, is the summation of all the root length—mm), seminal length (the total length of seminal roots—mm), lateral length (the total length of lateral roots—mm), mean seminal length (ASL, is the mean value of the total length of the seminal roots—mm), mean lateral length (mean value of the total length of lateral roots—mm), seminal count (PC, is the number of seminal roots), lateral count (number of lateral roots), mean seminal count (mean value of the total number of seminal roots), mean lateral count (mean value of the total number of lateral roots), average seminal emergence angle (measurement of emergence angle of the seminal roots—degrees), average lateral emergence angle (measurement of emergence angle of the lateral roots—degrees), average seminal tip angle (mean value of the measurement of angle in the seminal root tips—degrees), average lateral tip angle (mean value of the measurement of angle in the lateral root tips—degrees), root tip angle (the measurement of angle in the seminal root tips—degrees), maximum width (MW, is the furthermost width of the root system along horizontal axis—mm), maximum depth (MD, is the furthermost depth of the root system along vertical axis—mm), width–depth ratio (WDR, is the ratio of the maximum width to the maximum depth of the root system), centroid (the coordinates of the center of mass of root system along the horizontal, Cen_X and vertical axes, Cen_Y—mm), convex hull area (CHA, is the area of the smallest convex polygon covering the boundaries of the root system—mm$^2$), and tortuosity (the average curvature of the seminal roots).

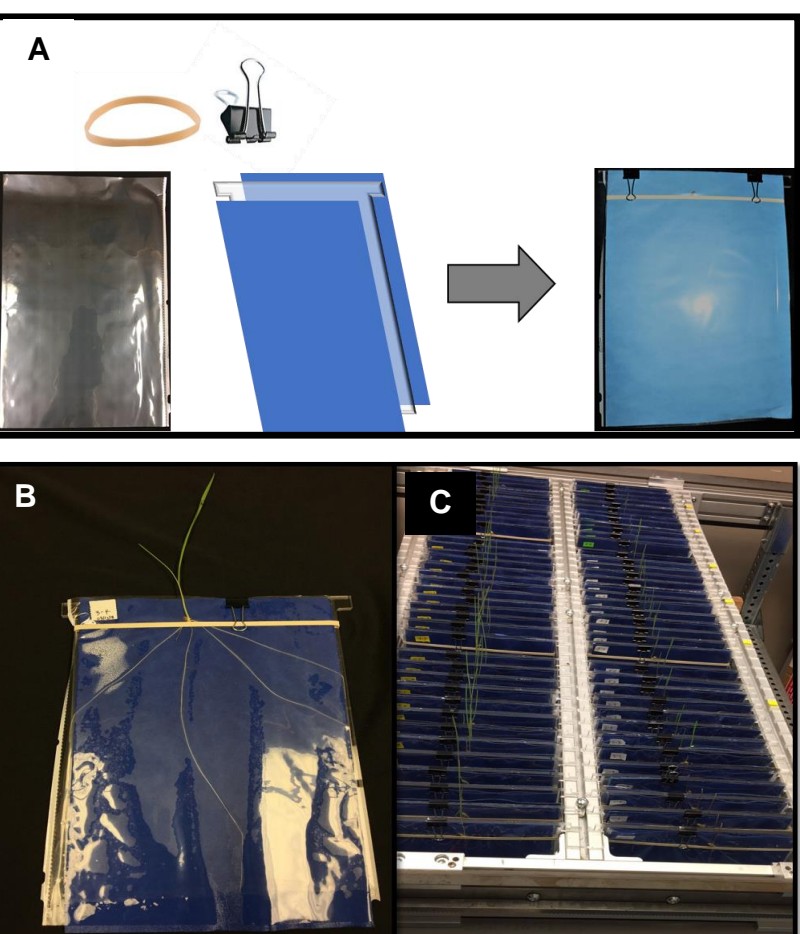

**Figure 4.** A customized high-throughput seedling root phenotyping platform, showing the growth assembly. The 2-D growth system, showing the growth pouch on one side. (**A**) The growth paper was inserted within the cover sheet that has had the bottom end removed. A rubber band and binder hold two germination pouches firmly in place to the acrylic plate. (**B**) The germinated seed shows the RSA of the wheat seedling at the two-leaf stage. (**C**) An assembled 2-D growth system showing growth pouches hanging from a metal frame.

Statistical analysis of the results obtained from RootNav was processed and analyzed using IBM SPSS Statistics for Windows, v25.0 (IBM Corp., NY, USA). The results obtained were expressed as mean values for each parental line for each trait. Analysis of variance (ANOVA) was applied to compare the means. Based on the outcome of the ANOVA on all data, Tukey HSD post hoc analysis was performed to separate the means.

Spearman rank correlation coefficients ($\varrho$) was used to determine associations between measured traits. Data analysis and visualization of the mixed model was performed using R software Version 3.4.3.

## 3. Results

The root of the wheat seedlings grew freely along the airspace between the clear propylene sheet and the moistened blue absorbent growth paper without growing into the paper. This allowed for the capturing of clearly distinguishable root images from the blue germination paper. RootNav software (1.8.1) was used to extract the quantification of RSA traits from the total root images of 312 seedlings that were captured 7 days after planting.

### 3.1. Frequency Distribution of Germination Potential and Measured Root Traits

The germination potential of each accessions is shown in Figure 5. The average germination rate of hexaploid was 9.4% higher than the tetraploid wheat accessions. The frequency distribution of the germination potential showed that 85.3% of all accessions exceeded a 90% germination rate (Supplementary Figure S1).

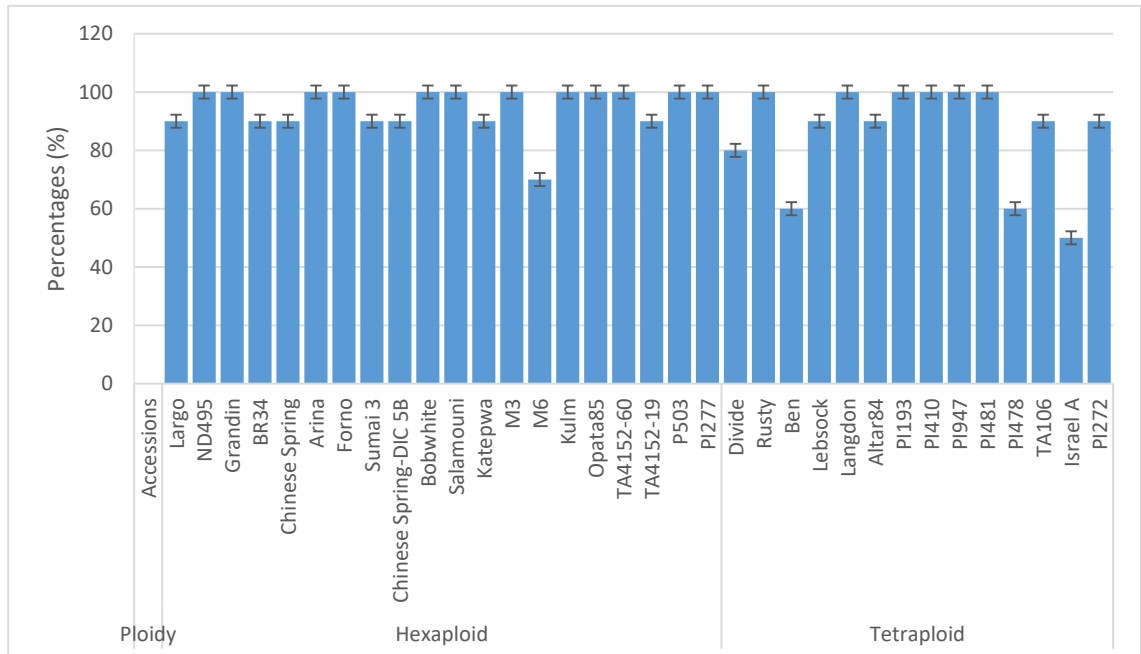

**Figure 5.** Germination potential for each accession.

The frequency histograms of the measured root traits for the 34 accessions are shown in Supplementary Figure S2. There was a strong correlation (0.8) between observed traits of total seminal root length and convex hull area. The average seminal length was strongly correlated (0.8) with maximum depth and centroid, while maximum width highly correlated (0.9) with width–depth ratio and convex hull area. The maximum depth also showed a high correlation (0.9) with a centroid (Figure 6).

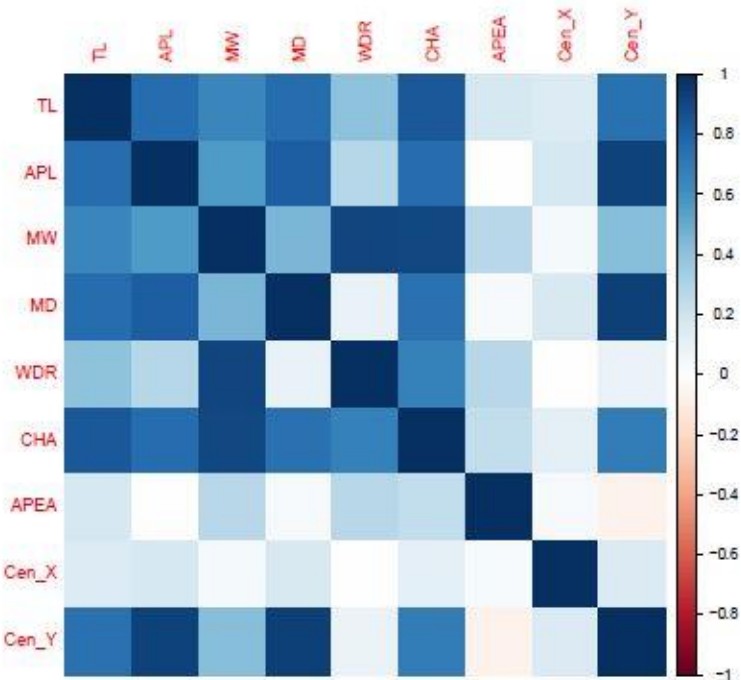

**Figure 6.** Correlation matrix of the measured root traits.

### 3.2. The Hexaploid Wheat Accessions

The non-destructive measurements of the RSA roots in Table 2 showed that the mean total length of Salamouni, Katepwa, Kulm, Opata85, TA60, Grandin, P503, Arina, Forno, Sumai3, and Chinese Spring-DIC 5B were significantly longer compared with Largo, which was used as the reference, by 0.9, 1.3, 1.9, 2.1, 1.5, 1.6, 0.9, 1.3, and 1.4 times, respectively. The average seminal length of Kulm, Opata85, TA60, Grandin, P503, Arina, and Forno compared to Largo were significantly longer, by 1.0, 1.0, 1.2, 0.8 1.0, 1.0, and 1.2 times, respectively. The mean count of the seminal root of Katepwa, Kulm, Opata85, and Grandin was significantly higher compared with Largo, by 0.5, 0.4, 0.5 and 0.4, times respectively. The mean maximum width showed that Kulm, Opata85, and Grandin were significantly larger compared with Largo, by 2.2, 1.4 and 1.7 times, respectively. The maximum depth of Kulm, Opata85, TA60, Grandin, P503, Arina, Forno, Sumai3, and Chinese Spring-DIC 5B were significantly greater compared with Largo, by 0.7, 0.8, 0.8, 0.6, 0.7, 0.7, 1.0, 0.8, and 0.8 times, respectively. The width to depth ratio of Kulm was significantly larger compared with Largo, by 0.9 times. The mean convex hull area of Kulm, Opata85, TA60, Grandin, and Arina were significantly larger compared to Largo, by 5.3, 4.8, 3.3, 4.1 and 3.1 times, respectively. The vertical coordinate of the centroid showed that Kulm, Opata85, TA60, TA19, P503, Arina, Forno, Sumai3, and Chinese Spring-DIC 5B were significantly greater compared with Largo, by 0.8, 1.0, 1.2, 0.8, 1.0, 1.1, 1.5, 1.0, 0.9 times, respectively.

**Table 2.** Root system architecture traits measured in 19 wheat accessions (hexaploid). The bolded mean values showed level of significance at *p* < 0.05 compared to the reference accession (Largo). Each trait has a column representing the mean value of 10 replicates for each accession followed by their standard deviation.

| Accession | Total Length Mean | STD | Seminal Length Mean | STD | Seminal Count Mean | STD | Maximum Width Mean | STD | Maximum Depth Mean | STD | Width-Depth Ratio Mean | STD | Convex Hull Area Mean | STD | Seminal Emergence Angle Mean | STD | Centroid_X Mean | STD | Centroid_Y Mean | STD |
|---|---|---|---|---|---|---|---|---|---|---|---|---|---|---|---|---|---|---|---|---|
| Largo | 1407.054 | 444.8384 | 400.8567 | 127.1285 | 3.56 | 0.726 | 332.22 | 179.31 | 751.89 | 182 | 0.4569 | 0.26354 | 105875 | 66594.71 | 22.6833 | 7.0495 | −0.008 | 67.68921 | 193.0758 | 63.93469 |
| ND495 | 1991.549 | 383.8933 | 532.204 | 207.123 | 4 | 0.816 | 378.8 | 260.032 | 1017.9 | 187.838 | 0.3512 | 0.19509 | 188694.2 | 162351.9 | 29.39 | 9.58095 | 3.6871 | 43.52455 | 279.4124 | 100.8176 |
| Salamouni | **2628.581** | 1295.576 | 611.519 | 269.4623 | 4.1 | 1.449 | 416.2 | 256.94 | 944.5 | 319.004 | 0.4251 | 0.24228 | 226789.1 | 154967.4 | 23.224 | 27.50345 | 76.3194 | 87.27794 | 277.9139 | 125.8842 |
| Katepwa | **3251.233** | 668.9902 | 644.2344 | 221.368 | **5.22** | 0.833 | 572.44 | 285.073 | 1035.33 | 210.224 | 0.5466 | 0.23586 | 333568.1 | 213121.5 | 31.5122 | 15.02541 | 9.21678 | 111.8648 | 278.9432 | 89.18816 |
| M3 | 2051.728 | 955.6283 | 531.802 | 145.6446 | 3.7 | 1.059 | 224.3 | 148.072 | 989.8 | 271.476 | 0.2196 | 0.12853 | 126481.6 | 117850.5 | 19.818 | 13.55339 | −4.9703 | 51.40844 | 284.2637 | 93.55736 |
| M6 | 1701.343 | 876.2018 | 504.2471 | 196.2748 | 3.29 | 1.254 | 278.57 | 178.362 | 860.29 | 241.249 | 0.3064 | 0.19161 | 120445.6 | 85442.86 | 23.2343 | 23.46807 | 26.79043 | 50.84016 | 239.2451 | 95.57134 |
| Kulm | **4071.759** | 780.7794 | **800.521** | 166.8296 | **5** | 0.471 | **1075** | 240.575 | **1274.8** | 297.392 | **0.8693** | 0.21052 | **671934.4** | 282573.1 | 30.945 | 3.86432 | 34.0647 | 110.5345 | **354.779** | 87.14734 |
| Opata | **4372.535** | 947.7119 | **786.908** | 224.1664 | **5.3** | 0.675 | 803.7 | 329.924 | **1355.4** | 202.447 | 0.595 | 0.2199 | **611183.9** | 348976.6 | 23.772 | 6.56108 | 0.5184 | 83.79973 | **383.1734** | 75.31793 |
| TA60 | **3453.33** | 739.9634 | **872.307** | 305.9635 | 4.1 | 0.876 | 632.8 | 283.294 | **1351.1** | 293.192 | 0.4748 | 0.20016 | **451817.1** | 236567.4 | 19.944 | 7.12346 | 43.7792 | 88.52216 | **433.4055** | 133.3076 |
| TA19 | 2331.48 | 441.3384 | 680.5678 | 183.436 | 3.56 | 0.726 | 221.22 | 186.238 | 1052.78 | 147.466 | 0.2186 | 0.21618 | 116801.6 | 74324.72 | 20.8933 | 6.69294 | 1.45578 | 49.8508 | 356.9752 | 70.49687 |
| Grandin | **3649.392** | 689.1213 | **727.775** | 157.4818 | **5.1** | 0.738 | **902.9** | 368.004 | **1177.09** | 294.013 | 0.803 | 0.41375 | **536945.6** | 253719 | 39.773 | 12.66347 | 9.7508 | 60.9755 | 334.9114 | 75.75706 |
| P503 | **2635.32** | 451.2482 | **815.313** | 167.5469 | 3.3 | 0.675 | 567.6 | 341.986 | **1288.2** | 244.35 | 0.4742 | 0.33644 | 336351.3 | 164600.2 | 35.085 | 18.39818 | 28.2041 | 62.06401 | **390.4914** | 94.16569 |
| BR34 | 2486.309 | 621.315 | 625.36 | 193.1676 | 4 | 0.866 | 505.11 | 162.079 | 1114.33 | 325.571 | 0.4858 | 0.17914 | 277661.6 | 117046.7 | 25.4289 | 7.82762 | 44.02322 | 37.9747 | 322.6331 | 110.6642 |
| CSpring | 2526.974 | 787.4891 | 599.1433 | 226.2535 | 4.33 | 1.225 | 366.78 | 197.488 | 1106.89 | 265.426 | 0.3302 | 0.16149 | 212942.8 | 131228.1 | 35.7667 | 6.43028 | 1.78556 | 73.44133 | 297.4512 | 127.6352 |
| Arina | **3211.019** | 799.5607 | **818.916** | 197.3953 | 4 | 0.816 | 623.2 | 283.085 | **1307.3** | 213.307 | 0.491 | 0.23865 | **430168.5** | 282573.1 | 185150.7 | 24.287 | 10.61435 | 70.9476 | 114.6787 | **403.6996** | 86.95764 |
| Forno | **2856.862** | 514.3885 | **891.16** | 116.4221 | 3.2 | 0.422 | 302.1 | 170.187 | **1499.6** | 281.496 | 0.2054 | 0.10853 | 246348 | 136554.2 | 24.115 | 15.79232 | 62.0981 | 91.38127 | **489.6822** | 94.50695 |
| Sumai3 | **3292.031** | 700.3864 | 700.5822 | 186.8713 | 4.78 | 0.667 | 332.11 | 130.143 | **1330.11** | 191.472 | 0.2538 | 0.10798 | 252744.3 | 128540.6 | 19.75 | 10.36522 | 8.46689 | 64.34472 | **394.5089** | 85.76959 |
| CSpringDIC | **3409.904** | 589.5194 | 704.0922 | 127.4161 | 4.89 | 0.601 | 532.78 | 243.404 | **1327.78** | 172.431 | 0.3948 | 0.16589 | 401419.2 | 188296.1 | 23.9867 | 10.9366 | 26.17744 | 51.21258 | **373.2856** | 67.95206 |
| Bobwhite | 2354.442 | 708.3413 | 723.972 | 213.3244 | 3.3 | 0.675 | 452 | 333.689 | 1155.6 | 244.926 | 0.3768 | 0.26688 | 248701.9 | 195832.2 | 17.424 | 7.47542 | 33.4653 | 65.00261 | 346.2481 | 85.93454 |

### 3.3. The Tetraploid Wheat Accessions

Accessions Langdon, PI 193883, PI 41025, PI 94749, and PI 272 were significantly higher in mean total length compared with Rusty (which was used as a reference) by 1.4, 1.6, 1.6, 1.8, and 1.4 times, respectively as shown in Table 3. For mean seminal length, Lebsock, PI 193883, PI 41025, and PI 94749 showed a significantly longer seminal root, with 1.5, 1.4, 1.6, and 1.5 times more than Rusty.

For the mean maximum width, PI 277 showed a significant difference, increasing 1.7 times more than Rusty. For mean maximum depth, PI 193, PI 410 and PI272 showed a significant difference, increasing 0.9, 0.8, and 0.8 times more than Rusty, respectively. For the mean width to depth ratio, the PI 277 showed a significant increase of 1.1 times more than Rusty (Table 3).

In other measured root trait, the mean convex hull area of PI 193883 significantly increased by 3.1 times when compared to Rusty. For centroid_Y, Lebsock, PI 193, PI 410 and Israel showed a significant difference, increasing 1.4, 1.6, 1.5 and 1.6 times respectively.

**Table 3.** Root system architecture traits measured in 15 wheat accessions (tetraploid). The bolded mean values showed level of significance at $p < 0.05$ compared to the reference accession (Rusty). Each trait has a column representing the mean value of 10 replicates for each accession followed by their standard deviation.

| Accession | Total Length Mean | STD | Seminal Length Mean | STD | Seminal Count Mean | STD | Maximum Width Mean | STD | Maximum Depth Mean | STD | Width-Depth Ratio Mean | STD | Convex Hull Area Mean | STD | Seminal Emergence Angle Mean | STD | Centroid_X Mean | STD | Centroid_Y Mean | STD |
|---|---|---|---|---|---|---|---|---|---|---|---|---|---|---|---|---|---|---|---|---|
| Divide | 2451.268 | 1990.837 | 556.8063 | 377.8616 | 3.75 | 1.909 | 179.5 | 183.808 | 968 | 467.962 | 0.163 | 0.11099 | 132953.1 | 141900.5 | 24.745 | 11.29471 | 62.338 | 54.1888 | 285.8023 | 183.9844 |
| Rusty | 1390.807 | 1255.677 | 314.033 | 226.8649 | 3.9 | 1.101 | 289.9 | 323.018 | 668.2 | 296.794 | 0.355 | 0.30589 | 117461.6 | 178758.6 | 19.006 | 10.84459 | 1.5814 | 46.94676 | 148.5815 | 108.4252 |
| Ben | 3226.34 | 1606.199 | 613.275 | 250.9193 | 4.67 | 1.862 | 325.5 | 176.039 | 1041.17 | 364.806 | 0.2894 | 0.17678 | 192996.1 | 100061.9 | 17.0567 | 6.29637 | −7.84817 | 31.76269 | 322.1412 | 136.3388 |
| Lebsock | 2709.894 | 1156.688 | **782.76** | 384.2838 | 3.56 | 1.13 | 352.78 | 215.12 | 1064.44 | 330.105 | 0.3325 | 0.18338 | 234695.5 | 203638.2 | 12.5878 | 12.2572 | 82.69922 | 67.46444 | **361.4402** | 171.0809 |
| Langdon | **3344.535** | 910.3358 | 647.958 | 140.2428 | 5.1 | 0.568 | 541.6 | 228.824 | 1119.8 | 198.192 | 0.4879 | 0.19377 | 319992.9 | 119248.3 | 31.939 | 15.33947 | −9.1897 | 60.98329 | 328.3687 | 72.69708 |
| Altar | 2509.834 | 1114.787 | 529.8867 | 216.3208 | 4.78 | 1.202 | 559.33 | 327.655 | 951.56 | 284.677 | 0.5769 | 0.27336 | 261532.6 | 218210.9 | 25.2367 | 12.27078 | 3.87411 | 44.27495 | 230.5439 | 106.7623 |
| PI193 | **3673.169** | 1406.628 | **745.731** | 256.8887 | 4.9 | 0.738 | 589.9 | 298.531 | **1266** | 337.197 | 0.4442 | 0.17098 | **484674.4** | 318533.1 | 25.628 | 12.4758 | −71.1325 | 67.57832 | **385.4912** | 132.6479 |
| PI410 | **3583.227** | 1531.398 | **814.01** | 338.0744 | 4.5 | 1.08 | 494.5 | 248.758 | **1211.7** | 369.315 | 0.4068 | 0.14769 | 351557.5 | 262529.1 | 15.345 | 6.84556 | 95.9932 | 142.1959 | **373.7213** | 141.8784 |
| PI947 | **3931.771** | 1070.18 | **772.071** | 210.8183 | 5.1 | 0.568 | 647.8 | 167.364 | 1125.7 | 280.126 | 0.6101 | 0.23353 | 389343.7 | 188785.1 | 34.931 | 8.18785 | 54.9579 | 125.2904 | 334.0316 | 96.79933 |
| PI481 | 1971.452 | 905.4402 | 707.311 | 291.7558 | 2.8 | 0.422 | 283.2 | 176.585 | 1045.6 | 315.264 | 0.2587 | 0.1009 | 162980 | 161573.2 | 17.845 | 13.14568 | −1.4536 | 44.00917 | 340.9898 | 134.9614 |
| PI478 | 1060.158 | 414.0258 | 449.97 | 99.81476 | 2.33 | 0.816 | 230 | 138.466 | 801.83 | 138.077 | 0.2761 | 0.16724 | 75115.56 | 49265.76 | 15.71 | 9.12903 | −1.32917 | 29.95209 | 214.8288 | 41.66894 |
| TA106 | 1789.02 | 682.8475 | 596.3411 | 227.6161 | 3 | 0 | 515 | 274.868 | 944.22 | 258.074 | 0.5188 | 0.18626 | 226387.3 | 185876.3 | 17.5533 | 7.23385 | −3.023 | 18.78727 | 275.3341 | 94.02308 |
| Israel | 2400.58 | 1152.096 | 814.988 | 352.7714 | 2.8 | 0.447 | 552.8 | 345.331 | 1215.6 | 395.11 | 0.4045 | 0.19579 | 331445.6 | 247329.2 | 18.316 | 9.29823 | 7.3922 | 39.28395 | **391.6284** | 161.1772 |
| PI277 | 2494.331 | 756.1297 | 586.835 | 250.0628 | 4.4 | 0.699 | **769.1** | 268.301 | 1056.8 | 313.966 | **0.7469** | 0.24797 | 388875.3 | 231645.1 | 27.434 | 9.31469 | 52.6313 | 47.89209 | 289.165 | 126.5571 |
| PI272 | **3406.442** | 1182.352 | 726.5822 | 217.0046 | 4.67 | 0.707 | 623 | 258.412 | **1232.33** | 278.285 | 0.5107 | 0.18407 | 418640.8 | 225100.3 | 26.1411 | 12.37334 | 62.85044 | 61.72506 | 342.3118 | 119.2106 |

## 4. Discussion

The root phenotyping pipeline, examined in this study using a germination paper-based moisture replacement system, allowed the measurement of important root architectural traits to be collected in an efficient, low-cost, and high-throughput fashion.

### 4.1. The Benefit of the Root System Size

The root system size is the representation of the total root length, seminal count, and the convex hull area. In previous studies, these traits have been positively associated with each other as well as with the grain yield of wheat in the field [22,23]. We also found a significant correlation between the total root length, maximum depth and the convex hull area in this study. A higher total root length does not always signify a deeper root growth, as the total length of seminal roots is the result of many seminal roots. Therefore, the addition of another pair of seminal roots to the root system results in a longer seminal root length as well, but not automatically in a deeper root system. Therefore, this results in an average seminal root length (defined as the total seminal length divided by the seminal root count), a preferable trait to allow root growth than total length, as a deeper root system may help to access deep water and mobile nutrients. In addition, the average seminal root length trait showed a significant correlation with both total root length and the convex hull area, which agrees with previous findings [32] that suggested that deeper penetration of the soil by seedling roots may result in better access to mobile soil nutrients and early plant establishment. The total root length and the average seminal root length had strong associations with the centroid_Y (vertical axis), which is suggested to be responsible for the aboveground vigor and root depth of the plant [12]. Based on our study, the use of average seminal length traits to assess variations in the RSA of wheat seedling is recommended because of its ability to delineate the performance of most of the wheat accessions, and at the same time showing a strong correlation with the next three important traits: total length, maximum depth and convex hull area.

### 4.2. Kulm and Opata85 May be Useful for RSA Improvement in Hexaploid Wheat

Kulm is a hard-red spring wheat (HRSW) developed at North Dakota State University, Fargo, ND. In our study, Kulm performed better than all the other accessions we examined under the same environmental conditions. Kulm had a higher mean total length, mean average primary length, seminal count and convex hull area, making it a suitable candidate for breeding a larger root system and greater spatial distribution. Kulm has been used in previous studies as a parental line for inbred line developments [33,34] with a report affirming its higher yield. Although Kulm has been found susceptible to some wheat pathogens, like septoria tritici blotch (STB) [34], *septoria nodorum* blotch (SNB) and tan spot [35], it remains a good candidate for the selection of grain-end-use quality [33].

Opata 85 is a commercial spring wheat cultivar developed at International Maize and Wheat Improvement Center (CIMMYT), Mexico [36]. In our study, the next best accession after Kulm was Opata85, as it produced more roots and an overall architecture that allowed it to occupy a greater root area. These traits make Opata 85 a suitable breeding candidate for larger root development and improvements in abiotic stress resistance. Opata 85 has been used as a parental line for recombinant inbred lines (RILs) used to map yield traits [37], important agronomic traits [36], growth characters [38], water-logging tolerance in seed germination and seedling growth [39], and growth duration components [40].

### 4.3. The Significance of a Rapid Screening Pipeline for Measuring Seedling Root Traits

The high-throughput root phenotyping pipeline that was developed in this study revealed variation in seedling root traits of both hexaploid and tetraploid wheat accessions. The pipeline allowed us to examine the root system architecture of 340 wheat seedlings, using only one out of four sections of our metal scaffoldings, that were fitted with three solution tanks. Each section has the

capacity to fit 84 growth systems that allow the screening of approximately 168 seedlings for each assembly. The total capacity of the platform can allow phenotypic evaluation of 672 plants per run in the fixed temperature growth room within 10 days, and this includes the assembling of a 2-D growth system and image analysis. Acquisition of root images of 168 seedlings takes approximately 3.5 hours, while the semi-automated image analysis using open source software takes 1.5mins per image. This is slightly more time efficient than the method of Atkinson et al., [12] who reported ~2mins per image and ~5 mins per plant.

The cost of the 2D-growth system is ~ \$0.43 per plant with a reusable acrylic sheet of ~ \$3.30. The overall growth system assembly for the first time will cost \$4.20 with a recurring cost of \$0.90 per system. This is 81% lower than the average available market price of seed germination pouches.

Although different phenotyping systems based on germination paper have been reported in previous studies [15,32,33], the pipeline described in this study is similar to the pouch and wick hydroponic-based system [12]. However, the pipeline in our study was enhanced by adding vapor sterilization of the seeds; positioning the wheat seeds at a strategic angle that improved root images; growing two (2) plants per growth system; and utilization of separable solution tanks that can hold up to 4L of nutrient solution and 28 growth systems. The advantage of this type of solution tank for further investigation is that root response to abiotic stress and different nutrient regimes [41] can be assessed by varying solution constituent. Recently, Shorinola et al. [42] implemented this phenotyping pipeline to conduct a forward genetic screening for variation analysis of seminal root in a Cadenza mutant population.

### 4.4. Future Work

SHW lines have become valuable resources for the genetic improvement of common wheat cultivars [29]. The findings of the variation analysis from this study will allow us to investigate segregating mapping populations that will include the RILs of M3 and Kulm; and M6 and Opata85. M3 was developed at CIMMYT, Mexico whereas Kulm was developed at North Dakota State University, Fargo, ND. These hexaploids are both spring type, with M3 being a synthetic hexaploid while Kulm is a hard-red spring wheat [34]. The associative mapping population that resulted from the crossing of these two lines (Kulm × M3) resulted in the 105 RILs that will be used in further studies of the hexaploid lines. Additionally, 114 RILs resulting from the hexaploid mapping population of Opata85 × M6 and chromosome substitution lines involving PI 478742, a tetraploid (where individual pairs of chromosomes of wild emmer have been substituted for homologous pairs of chromosomes in background of Langdon durum), will be evaluated to identify the chromosome locations of loci responsible for the differences in RSA traits. Thereafter, molecular markers suitable for the marker-assisted selection of these traits will be developed.

### 5. Conclusions

In this study, we have studied RSA on 34 different wheat accessions at an early stage of plant development and were able to demonstrate its use in identifying accessions which perform better than others in some of the RSA characters. This study clearly possesses an advantage over the previously reported study because of its capacity to increase screening potential at early stages of plant development. This pipeline is also very simple and provides an opportunity for automation of acquisition of the RSA images (Step 3 of Figure 2) and screening platforms. The availability of mapping populations and high-resolution mapping data from these accessions provides an opportunity for utilizing this pipeline in identifying QTLs linked to RSA in populations segregating in RSA traits.

**Supplementary Materials:** The following are available online at http://www.mdpi.com/2073-4395/10/2/206/s1, Figure S1: Frequency distribution of the germination potentials (percentage) of the wheat accessions evaluated title, Figure S2: The frequency histograms of measured root traits.

**Author Contributions:** The method was conceived, designed, and coordinated by A.T. The experiment was performed by E.A., R.M. and W.M. The data were analyzed and results interpreted by E.A., E.A. wrote

the manuscript with input from J.F. and A.T. All authors have read and agreed to the published version of the manuscript.

**Funding:** This study was funded by USDA Evans-Allen grant number: 1005722.

**Acknowledgments:** We appreciate and thank the anonymous reviewers for their valuable inputs and comments that have improved the quality of this manuscript.

**Conflicts of Interest:** The authors declare no conflict of interest.

## Abbreviations

| | |
|---|---|
| TL | Total length; |
| APL | Average seminal length; |
| MW | Maximum width; |
| MD | Maximum depth; |
| WDR | Width-depth ratio; |
| CHA | Convex hull area; |
| APEA | Average seminal emergence area; |
| Cen_X | Horizontal coordinates of centroid; |
| Cen_Y | Vertical coordinates of centroid; |
| PC | Seminal count. |

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
