# Peer review of "Variation Analysis of Root System Development in Wheat Seedlings Using Root Phenotyping System"

_agronomy, doi:10.3390/agronomy10020206_

Round 1

Reviewer 1 Report

The manuscript entitled “Variation Analysis of Root System Development in Wheat Seedlings Using Root Phenotyping System ", written by Adeleke and co-authors, reports on introducing a very much needed phenotyping pipeline for root system architecture in wheat seedlings. Such work might be adapted in other cereals and plants as well. The authors described their methodology very well and in detail.

Here are my comments:

Line 55:

Figure 1 shows germination stage in wheat and more precisely Zadoks stages 05-09, where the first root appear to appearance of first leaf at the tip of coleoptile. I suggest authors change “two-leaf” stage to “germination stage” instead of changing figure 1, as there is no leaf in figure 1.

Line 61:

I recommend removing in “manipulate in”

Line 63:

I recommend changing “to” in “critical to optimizing” to “critical for optimizing”

Line 71:

I recommend removing “development of” in the beginning of line 71.

Line 115:

I recommend writing “Figures” instead of “Figure” and there is no A, B and C on figure 3. Please add A, B & C on Figure 3.

Line 123:

I recommend removing “-“from Figure 2-B

Table 1:

I recommend categorizing the table into hexaploid and tetraploid better by putting hexaploid accessions one after each other and the same for tetraploid accessions. This can improve the readability of table 2.

Figure 2:

I recommend improving Figure 2 by changing it to a horizontal (landscape) mode and by increasing the font size of the writings in the boxes and use lighter colors.

Figure 3:

Please add A, B and C to the picture so the reader can trace the description easier.

Figure 4:

In the caption, there is a description for Figure 4B section, but manuscript never referred to Figure 4B in the text.

Line 201:

This is the first time that authors mentioning “PL” in the text and there is no clue in the rest of manuscript what PL stands for. Please address this.

Figure 6-15:

The caption says Figures 8-15. Please change it to 6-15. There is no description on what stars mean on the bars. Please mention in the caption that stars are showing significance difference compared to the reference accession in each category, mentioning level of significance as well. Increase resolution for all graphs in figure 6-15 and increase the font size in each axis. The authors did not mention anything about the results of Figures (15A and 15B) for mean centroid X in the text.

Line 244:

Add (Figure 15B) after respectively.

Author Response

Thank you very much for your comments and suggestions, they have gone a long way to make our manuscript a better version.

Please find in the attached file our response to the major comments:

Reviewer 2 Report

The authors present a method for characterizing root system architecture in wheat seedlings. The manuscript was well written and the method will be of interest to the scientific community. However, the results/main findings of the experiment remain somewhat unclear.

Comments:

The description of the method was very good and will be useful to the community. I think that the experimental results of the study will also be useful, but I found the main findings or trends in the data unclear. What is the biological question and do the trends in the data help answer that question? Is there a trend across the ploidy levels, do the SHW appear different than elite cultivars or the spelt, domesticated vs. undomesticated, etc. (I am not expecting the authors to answer these questions specifically, but I was not able to easily answer these when looking at your results/figures)? Can the biological question and results be presented in a way to make the questions, main findings, and the significance of the results more clear? There was some discussion on two specific lines, Kulm and Opata85, but the overall value of the majority of the results could be better presented both in the figures and in the text. Perhaps it is just how the data is presented in the figures, and presenting them differently might improve their value. Many of the figures for the results are presented in oversized bar graphs that span many pages, it should be possible to summarize the data into single page figures or tables that make the trends more clear. Currently the scientific results are not presented in a way that makes the data trends clear. For example, Figure 5. Not all line names are shown. Perhaps if more space is needed for labels, the graph can be rotated. As another example, one of the figures was labeled 8-15 – I am assuming that this is Figures 6 (not 8) to Figure 15? I did not like this figure. I found the numbering/labelling unusual and the patterns in the data unclear. For the labelling, perhaps just label the panels A-Z where left is hexaploid and right is tetraploid, or have a separate figure for hexaploids and tetraploids, or have them both in the same plot area and use different colored bars for ploidy level, SHW, and spelt, or wild vs. domesticated. The resolution of this figure was also too low to read any of the labels. If the lines remain in the same order, perhaps only one label is needed. For the data trend, given that this is the core results from the manuscript, I think it would be important that this data be presented as a main figure or raw values in a table and be carefully presented to highlight the patterns in the data – ideally on a single page or as separate figures aimed at each data trend. The figure should also have a more complete caption.

I liked Supplemental Figure S3 and suggest it be a main figure. I suggest adding percent germination to the correlation analysis – seed health may be impacting both germination and RSA.

There was not much discussion on durums or spelt in the Discussion section.

Line 10: sentence is missing words or is poorly structured. establishment of what?

Line 18: missing a space before days.

Line 40: bad comma use

Line 61 manipulate the (remove ‘in’)

Line 96: vs is an abbreviation needs a period after (vs.)

Line 96: It says supplemental, but I think that this is a main table.

Line 96. It may be worth describing what synthetic hexaploid wheats are

Line 146: comma after ‘background’

Line 184: missing a period after ‘paper’

Line 293: space before ‘days’

Line 295 and elsewhere: space before ‘mins’

Line 310: ‘a Cadenza mutant population’

Line 319: incomplete thought? ‘the hexaploid’

Line 319: space before RILs

Line 331: Capitalize ‘figure’

Author Response

(The authors gave the same response as above.)
